# Preparation and Embedding Characterization of Hydroxypropyl-β-cyclodextrin/Menthyl Acetate Microcapsules with Enhanced Stability

**DOI:** 10.3390/pharmaceutics15071979

**Published:** 2023-07-19

**Authors:** Xiaoqing Huang, Honghui Guo, Quanling Xie, Wenhui Jin, Runying Zeng, Zhuan Hong, Yiping Zhang, Yucang Zhang

**Affiliations:** 1College of Marine Food and Biological Engineering, Jimei University, Xiamen 361021, China; 202011832006@jmu.edu.cn; 2Engineering Technology Innovation Center for the Development and Utilization of Marine Living Resources, Third Institute of Oceanography, Ministry of Natural Resources, Xiamen 361005, China; qlxie@tio.org.cn (Q.X.); whjin@tio.org.cn (W.J.); zeng@tio.org.cn (R.Z.); hzh@tio.org.cn (Z.H.); ypzhang@tio.org.cn (Y.Z.); 3Department of Marine Biology, Xiamen Ocean Vocational College, Xiamen 361100, China; 4Fujian Provincial Key Laboratory of Island Conservation and Development (Island Research Center, MNR), Pingtan 350400, China

**Keywords:** menthyl acetate, hydroxypropyl-β-cyclodextrin, microcapsule, gas chromatography, embedding rate, stability, thermodynamic parameter

## Abstract

Objective: Hydroxypropyl-β-cyclodextrin (HP-β-CD)/menthyl acetate (MA) microcapsules were developed to overcome the volatile and unstable defects of MA and improve the ease of use and storage. Methods: MA microcapsules were prepared via spray drying using HP-β-CD as the wall material. The embedding rate of MA microcapsules was determined through gas chromatography. The embedding characteristics were studied using phase solubility and nuclear magnetic resonance (NMR). The stability was characterized via differential scanning calorimetry (DSC) and the release and retention rates of MA microcapsules at different temperatures. Results: The embedding rate of HP-β-CD /MA microcapsules was 96.3%. The Gibbs free energy change, enthalpy change and entropy change of the embedding reaction between HP-β-CD and MA were all less than zero, indicating that the embedding process was a spontaneous exothermic reaction. NMR spectra showed that MA entered the cavity of HP-β-CD through the large opening end and interacted with the inner wall of the small opening end. DSC and the release and retention rates of MA microcapsules at different temperatures showed that the stability of MA was significantly enhanced after being embedded in HP-β-CD. Conclusion: The HP-β-CD/MA microcapsules are able to significantly improve the stability of MA and reduce the volatilization of MA.

## 1. Introduction

Menthyl acetate (MA) is the main component of peppermint oil and one of the simplest and most common menthol esters. MA has a unique minty, rose-like, cool smell and a refreshing effect [1,2]. It has the functions of promoting transdermal absorption; anti-inflammatory, analgesic, antifungal and antiviral properties; and has certain effects on the central nervous system, digestive system, reproductive system and respiratory system. Currently, it has been widely used in many fields [3,4], such as food, medicine, cosmetics, and daily necessities. However, MA has some disadvantages, such as instability, volatility and poor aqueous solubility, so its storage and use are limited to a certain extent, especially in a high-temperature environment, resulting in its low utilization and value loss. 

Microencapsulation technology has attracted much attention in recent years. Microencapsulation can enhance the stability [5,6] of a core material, eliminate undesirable taste or odor, control chemical reactions and molecular release, etc. [7,8,9]. Therefore, it is feasible to improve the heat resistance and stability of MA through microencapsulation. In addition, there are few studies on the microencapsulation of MA, so selecting a hydrophobic carrier for MA is very significant. Cyclodextrin has hydrophilic properties externally and hydrophobic properties internally and has low toxicity, high stability and low cost. It is widely used in the stabilization, masking or controlled release [10] of hydrophobic substances. Cyclodextrin-based drug delivery systems can enhance the permeability of biological barriers, such as skin and mucosa; improve the disposition of drugs in the therapeutic target; reduce the side effects of drugs; and effectively improve the bioavailability of drugs for different routes of administration [11]. Among the types of cyclodextrins, β-cyclodextrin (β-CD) is the most widely used due to its size, allowing it to properly host several sizes of molecules, and due to its lack of toxicity when orally administered. However, it has low gastrointestinal absorption compared to other cyclodextrin derivatives and may cause renal toxicity when administered parenterally. Hydroxypropyl-β-cyclodextrin (HP-β-CD) is a derivative of cyclodextrin. The introduction of hydroxypropyl in HP-β-CD broke the cyclic hydrogen bond within the molecules of β-CD, while maintaining the β-CD cavity to overcome the shortcomings of the low aqueous solubility of β-CD. The aqueous solubility of HP-β-CD is greater than 50%, which is 27 times [12,13] higher than that of β-CD, and the embedding capacity of HP-β-CD is stronger than that of β-CD [14]. Compared with β-CD, HP-β-CD can further improve the drug encapsulation rate, enhance drug stability and have better tolerance to oral administration, intravenous injection, intradermal injection and non-gastrointestinal administration. Animal experiments show that HP-β-CD has good safety, non-toxic side effects and low hemolysis effects on animals; for instance, hemolysis is nine times lower than β-CD and fifteen times lower than dimethyl-β-cyclodextrin (DM-β-CD). Therefore, HP-β-CD has a good application prospect in the medicine and food fields.

The common methods to determine the content of volatile oil in the inclusion are steam distillation [15,16], constant temperature weight loss method [17] and ultraviolet spectrophotometry [18,19], among which the steam distillation pretreatment is more complicated, the determination time is long and the efficiency is low. The constant temperature weight loss method results in significant errors and poor repeatability, while ultraviolet spectrophotometry is complicated and consumes a large number of reagents and the cost is high [20]. It is difficult for the above methods to meet the requirements of the rapid determination of MA in microcapsules. The determination of volatile oil via gas chromatography has the advantages of simple operation and high sensitivity. Therefore, this paper uses and optimizes gas chromatography [21,22,23] to establish an efficient detection method suitable for MA in microcapsules. On this basis, HP-β-CD/MA microcapsules with enhanced MA solubility and stability were prepared via the spray drying method. The embedding properties of HP-β-CD/MA microcapsules were fully characterized to explore the host–guest interactions. This provides an important basis for the broad application of MA and HP-β-CD.

## 2. Materials and Methods

### 2.1. Materials

Ethyl acetate (chromatographic grade) and HP-β-CD (purity > 97%) were purchased from Shanghai Aladdin Biochemical Technology Co., Ltd. (Shanghai, China). Ethanol (chromatographic grade), and MA (purity > 99%) were purchased from Aldrich. 

### 2.2. Preparation of Microcapsules

A total of 12 g of MA and 2.4 g of Tween 60 was added into 40 mL pure water and was homogenized using a PB100 handheld homogenizer (Shanghai, China) at a high speed of 5000–6000 rpm for 3–5 min to obtain a homogeneous emulsion. A total of 24 g of HP-β-CD was added into 200 mL pure water and stirred sufficiently until it was clarified and transparent in order to obtain an aqueous solution of the wall material. The homogenized emulsion was added into the aqueous solution of the wall material in a constant temperature water bath (50~60 °C) and stirred for 2 h. Finally, HP-β-CD/MA microcapsules were obtained via spray drying (SY6000, Shanghai, China). The air inlet temperature was 130 °C, and the air outlet temperature was 100 °C. 

### 2.3. Determination of MA by Gas Chromatography

#### 2.3.1. Gas Chromatographic Conditions 

MA was characterized by 7890A gas chromatography (Agilent Technologies, Santa Clara, CA, USA). Detector: FID; Column: HP-5 capillary column 30 m × 0.250 mm × 0.25 µm; carrier gas: N_2_; flow rate: 1.00 mL/min; injection volume: 0.8 µL; temperature: initial temperature 60 °C, at the rate of 10 °C/min, heating up to 200 °C; detector temperature 280 °C; injection port 280 °C; shunt ratio 10:1.

The standard method [24], GB28337-2012: “National Food Safety Standard for Food Additive Menthylate Acetate”, was used as a control method to detect MA content. Chromatographic conditions are as follows. Detector: FID; column: HP-5 capillary column 30 m × 0.320 mm × 0.25 μm; carrier gas: N_2_; Flow rate: 1.00 mL/min; injection volume: 0.2 μL; temperature: 220 °C constant temperature; detector temperature 250 °C; inlet 250 °C. The shunt ratio was 10:1.

#### 2.3.2. Sample Solution Preparation

A total of 0.1 g of HP-β-CD/MA microcapsule sample was added into 5 mL ethanol and ethyl acetate, respectively. A total of1 mL of the sample solution in ethanol was filtered through a 0.22 μL filter membrane and used as the test solution I. The sample solution in ethyl acetate was centrifuged at 3000 rpm for 2 min. Then, the supernatant was filtered through a 0.22 μL filter membrane and used as the test solution II. The experimental scheme of MA detection process is shown in Figure 1.

#### 2.3.3. Standard Curve

A total of 0.02 g of MA sample was dissolved with ethanol in a 10 mL volumetric bottle as reference liquid I. Similarly, 0.02 g of MA sample was dissolved with ethyl acetate in a 10 mL volumetric bottle as reference solution II. Then, 1000 μL, 500 μL, 100 μL, 10 μL and 1 μL of reference solutions I and II were, respectively, diluted with ethanol and ethyl acetate at a 10 mL constant volume. Then, 1 mL was filtered through a 0.22 μm filter membrane and injected for determination. Finally, the standard curve was plotted with the peak area as the ordinate and the concentration of MA as the abscissa.

#### 2.3.4. Precision Test

Precision refers to the degree of closeness between the results obtained by at least five repeated measurements of the same sample under specified test conditions. It is usually represented by relative standard deviation (RSD). A total of 2 mg/mL of reference solution I and reference solution II were injected 6 times to determine the peak area, and the RSD was obtained.

#### 2.3.5. Recovery Test

Recovery was determined via the sampling recovery method [25]. The ethanol solution and ethyl acetate solution containing 1 mg MA were accurately removed and added into the ethanol solution and ethyl acetate solution of the sample, respectively. According to the conditions of gas chromatography in Section 2.3.1, the content of MA was measured to calculate the average recovery, and the formula is as follows:(1)$$\mathrm{Recovery}=\frac{A_{1}}{A_{0}+X}\times 100\%$$
where A_1_ is the average content of MA when a known amount of MA is added to the sample, mg/mL; A_0_ is the initial average content of MA in the sample, mg/mL; and X is the known content of added MA, mg/mL. 

### 2.4. Determination of Embedding Rate

A total of 0.1 g of MA, HP-β-CD and HP-β-CD/MA microcapsule samples were dissolved in 5 mL with ethanol. A total of 1 mL of solution was filtered by 0.22 μm filter membrane, and samples were injected according to gas chromatography conditions in Section 2.3.1.

HP-β-CD and MA were dissolved in ethanol, and the total content of MA in the HP-β-CD/MA microcapsules was determined using ethanol as the solvent. Since MA was soluble in ethyl acetate, and HP-β-CD was insoluble in ethyl acetate, ethyl acetate could be used as a solvent to determine the unembedded MA content in the sample. The embedding rate of microcapsules was calculated according to the total content of MA in the microcapsule sample and the unembedded content of MA on the surface of the microcapsules. The formula is as follows: (2)$$\mathrm{Embedding}\;\mathrm{rate}/\%=\frac{W_{1}-W_{2}}{W_{1}}\times 100\%$$
where W_1_ is the total content of MA in microcapsules, g/g, and W_2_ is unembedded MA content on the surface of microcapsules, g/g. 

### 2.5. Thermodynamic Analysis

Phase solubility [26] was used to determine thermodynamic parameters. Amounts of 2, 4, 6, 8, and 10 mmol/L of HP-β-CD aqueous solutions were prepared, respectively. A supersaturated amount of MA was added into 5 mL of each of the above aqueous solutions. Then, they were oscillated at 120 r/min for 48 h under different temperature conditions (25 °C, 30 °C and 35 °C), so the solution could reach dissolution equilibrium. The supernatant was filtered through a 0.22 μm filter membrane, and 5 mL of the filtrate was taken and placed in a 50 mL centrifuge tube. A total of 10 mL of ethyl acetate was added for extraction, and 1 mL of the extract was taken and determined according to the chromatographic conditions in Section 2.3.1. With MA concentration (mmol/L) as the vertical coordinate and HP-β-CD concentration (mmol/L) as the horizontal coordinate, the phase solution diagram was plotted, and linear regression was performed to obtain the regression equation.

According to Higuchi and Connors’ theory, the stability constant K of the embedded solution of HP-β-CD and MA was calculated at different temperatures.
(3)$$K=\frac{a}{b(1-a)}$$
where a is the slope of the regression equation and b is the intercept of the regression equation. 

The enthalpy change (ΔH), entropy change (ΔS) and Gibbs free energy change (ΔG) were calculated according to the relationship between the stability constant K of the embedded solution at different temperatures and the temperature T, namely the Van ’t Hoff equation.
(4)$$\ln K=-\frac{\Delta H}{\mathrm{RT}}+\frac{\Delta S}{R}$$
(5)ΔG=ΔH−TΔS
where K is the stability constant of HP-β-CD and MA at different temperatures, L/mol, and R is the molar gas constant 8.314 J/mol·K.

### 2.6. Nuclear Magnetic Resonance Detection (NMR)

Avance III-400 MHz all-digital nuclear magnetic resonance spectrometer (Brooke Company, Karlsruhe, Germany) was used for detection [27]. A total of 20 mg microcapsules of MA, HP-β-CD/MA were weighed and dissolved in 1 mL DMSO organic solvent. Tetramethylsilane (TMS) was added, and the peak was set at 0.00 ppm chemical shift as the internal standard. The experimental temperature was set at 303 K, and the resonant frequency was 400 MHz. 

### 2.7. Differential Scanning Calorimetry (DSC)

The thermal properties of MA and HP-β-CD/MA microcapsules were determined using TAQ2000 (Waters, Framingham, MA, USA) differential scanning calorimeter [28]. A total of 4 mg of sample was placed onto the sample tray and heated from 30 °C to 300 °C at a rate of 10/min in the nitrogen environment. The relationship curve between the heat and temperature of the sample was obtained.

### 2.8. Stability Tests at Different Temperatures

#### 2.8.1. Stability Test at 100 °C

A total of 2 g of HP-β-CD/MA microcapsules were placed in a glass Petri dish and heated in an oven at 100 °C. A total of 0.1 g of sample was taken every 10 min, and the content of MA was measured according to the gas chromatographic conditions described in Section 2.3.1. The formula for calculating the release rate of MA in microcapsules is as follows [29].
(6)$$\mathrm{Release}\;\mathrm{rate}\left(\%\right)=\frac{m_{0}-m_{1}}{m_{0}}\times 100\%$$
where m_0_ is the initial content of MA in the sample, g/0.1 g, and m_1_ is the content of menthol acetate sampled during heating, g/0.1 g.

#### 2.8.2. Stability Test at Room Temperature 

A total of 2 g of MA and HP-β-CD/MA microcapsule samples were accurately weighed at room temperature and placed into a constant temperature and humidity box (temperature 25 °C humidity 60%). A total of 0.1 g of sample was taken daily, and the content of MA content was tested according to the gas chromatographic conditions in Section 2.3.1. The formula for calculating the retention rate of MA in microcapsules is as follows:(7)$$\mathrm{Retention}\;\mathrm{rate}\left(\%\right)=\frac{m_{1}}{m_{0}}\times 100\%$$
where m_1_ is the content of MA in the sample, g/0.1 g, and m_0_ is MA content in the sample initially, g/0.1 g. 

### 2.9. Statistical Analysis

All tests were performed in triplicate, and the results are presented as mean ± standard deviation. All the data were analyzed using OriginPro 2021(Version 9.8.0.200). 

## 3. Results and Discussion

### 3.1. Determination of MA

As can be seen from Figure 2A, the peak time of MA with ethanol as solvent was about 19.68 min. In Figure 2B, the peak time of MA with ethyl acetate as solvent was approximately 19.69 min. Both of the peak times were almost the same. Using ethanol as solvent, the regression equation was y = 1526.32x + 3.20 (R^2^ = 0.9999), indicating that MA had a good linear relationship in the concentration range of 0.002~2 mg/mL. The detection and quantitative limits were 0.83 μg/mL and 2.52 μg/mL, respectively. The standard curve was also plotted with ethyl acetate as the solvent, and the corresponding regression equation was y = 1580x + 15.65 (R^2^ = 0.9998), indicating that MA showed a good linear relationship in the concentration range of 0.002~2 mg/mL, and the detection and quantitation limits were 1.21 μg/mL and 3.8 μg/mL, respectively. The detection limit and quantitation limit of MA were 15.95 μg/mL and 52.63 μg/mL when ethanol was used as the solvent and 20.02 μg/mL and 66.67 μg/mL when ethyl acetate was used as the solvent for the determination of MA via the control method. In conclusion, this detection method is more sensitive than the control method. Both the control method and our method use gas chromatography, but the detection sensitivity is closely related to gas chromatography conditions, such as the capillary column diameter and operating temperature. Generally, programmed heating can optimize the separation effect to improve resolution and sensitivity [30]. The capillary column diameter of our method is significantly smaller than that of the control method, contributing to the high detection sensitivity due to the increased column efficiency.

Precision can reflect the accuracy of the detection method. When ethanol and ethyl acetate were used as solvents to detect MA, the RSD was 0.7% and 0.6%, respectively, which met the precision requirements in GB/T 27404-2008 “Physicochemical Testing of Food in the Scope of Laboratory Quality Control”. This demonstrates that the detection method can accurately determine the MA content.

The average recovery of MA in the ethanol solution was 99.8%, and the RSD was 1.3% (*n* = 6). The average recovery of MA in ethyl acetate solution was 100.3%, and the RSD was 0.8% (*n* = 6). The average recovery of MA in ethanol and ethyl acetate solutions of the control method was 99.2% and 99.8%. This method was consistent with the determination results of the control method, indicating that the recovery rate of this method was reasonable. It is able to meet the detection requirements of the embedding rate and the stability of MA microcapsules.

### 3.2. Embedding Rate of MA Microcapsules

In order to determine the influence of HP-β-CD on the gas phase detection results of MA, the gas chromatograms of MA, HP-β-CD/MA microcapsules and HP-β-CD were all detected. As can be seen from Figure 3, the gas chromatogram of both MA and HP-β-CD/MA microcapsules showed a peak, and the peak time of HP-β-CD/MA microcapsules was consistent with that of MA, while the gas chromatogram of HP-β-CD had no pronounced peak. This indicated that HP-β-CD had no interference in the determination of MA.

The test solutions I and II were injected six times, respectively. The average total content of MA was 0.347 g/g when ethanol was used as a solvent, and the average content of unembedded MA on the surface of the microcapsules was 0.0128 g/g when ethyl acetate was used as a solvent. According to formula (2), the embedding rate of HP-β-CD to MA was about 96.3%. 

### 3.3. Thermodynamic Parameters

Figure 4 is the phase solubility diagram of HP-β-CD and MA at different temperatures. It can be seen that the solubility of MA increases with the increase in the concentration of HP-β-CD at different temperatures, indicating the formation of soluble inclusions. Table 1 shows the linear regression equation and the stability constant of HP-β-CD-MA at different temperatures. The stability constant is an important parameter to measure the stability of the inclusion of cyclodextrin, which is positively correlated with the inclusion effect and thermal stability. The larger the stability constant, the better the inclusion effect and the more stable the inclusion substance will be. The increase in temperature often leads to the instability of the inclusion and the decrease in the stability constant. As seen from Table 1, the stability constant of HP-β-CD and MA decreases with the increase in temperature, and the embedding equilibrium moves toward dissociation, indicating that the embedding process of HP-β-CD and MA may be exothermic. The embedding process is a complex dynamic process, which is not only related to environmental factors, such as temperature, but is also closely related [31] to the shape, geometric size and molecular configuration of guest molecules. Table 2 shows the thermodynamic parameters of the embedding reaction between HP-β-CD and MA. It can be seen from Table 2 that the enthalpy change (ΔH), entropy change (ΔS) and Gibbs free energy (ΔG) in the embedding reaction between MA and HP-β-CD are all negative values, which further proves that the embedding process is an exothermic reaction. It is beneficial to the formation of HP-β-CD/MA microcapsules, and the steric hindrance of the inner cavity of HP-β-CD has an excellent protective effect on MA [32,33].

### 3.4. NMR Analysis

^1^H NMR spectroscopy is one of the most direct pieces of evidence for determining the intermolecular interactions in inclusion based on the chemical shifts of the subject and guest molecules [34]. H-1, H-2, H-4 and methylene H-6 are protons located on the outer surface of the cyclodextrin. In contrast, H-3 and H-5 are located on the inner surface of the cyclodextrin cavity, with the H-3 proton located on the large end and the H-5 proton near the small end [35]. Figure 5 shows the ^1^H NMR of MA, HP-β-CD and HP-β-CD/MA microcapsules in DMSO, and the insets show the proposed structure of the MA, HP-β-CD and HP-β-CD/MA microcapsule. As shown in Figure 5a, the characteristic peaks of MA appeared at the chemical shifts of 4.569, 1.986, 1.842, 1.631, 1.327, 0.866 and 0.716, corresponding to C_1_-H, C_13_-H, C_2,6_-H, C_3,7_-H, C_4,5_-H, C_8,9_-H and C_10_-H, respectively. In HP-β-CD/MA microcapsules, the characteristic peak of HP-β-CD and the characteristic peak of MA appeared, as shown in Figure 5c, indicating the formation of HP-β-CD/MA microcapsules. The chemical shift of the proton peak of the main HP-β-CD was changed after embedding. H-1 (4.835) and H-6 (3.753) on the outer part of the cavity did not change, while the chemical shift of H-3 in the cavity moved slightly to the high field from 3.615 to 3.613, and H-5 showed an apparent high field shift, moving from 3.555 to 3.534. The chemical shifts of C_1_-H, C_2,6_-H, C_3,7_-H, C_4,5_-H and C_10_-H of MA were −0.003, −0.001, −0.004, 0.001 and −0.002, respectively, indicating that it was inserted into the cavity, where C_13_-H and C_8,9_-H without a chemical shift may be exposed to the outside of the cavity. Since the diameter of the large end of the hydrophobic cavity of the HP-β-CD molecule is 0.78 nm, and the three-dimensional size of the MA molecule is 1.188 nm × 0.773 nm × 0.571 nm, it can be seen that the width of the MA molecule matches the size of the cyclodextrin cavity [36]. In conclusion, MA was not adsorbed onto the outer wall of the cavity of HP-β-CD. HP-β-CD/MA microcapsules were formed by MA entering the cavity from the large end of HP-β-CD molecules and embedding near the small end to interact with the inner wall of the cavity.

### 3.5. DSC Analysis

The DSC diagrams of MA, HP-β-CD and HP-β-CD/MA microcapsules are shown in Figure 6. MA presents an obvious endothermic state below 159 °C, which is consistent with its volatility. An endothermic peak of HP-β-CD appears at about 72 °C. HP-β-CD/MA microcapsules have a broad endothermic peak in the temperature range of 30~100 °C and the peak appears at 66 °C, which is mainly the water loss peak of the microcapsules. Compared to HP-β-CD, the endothermic peak of HP-β-CD/MA microcapsules is shifted. It is speculated that the binding of water molecules to the HP-β-CD cavity is reduced after MA occupies the HP-β-CD cavity. At the same time, the characteristic endothermic peak of MA at 159 °C disappears. A small endothermic peak appears at 226 °C, which is the melting temperature of HP-β-CD/MA microcapsules, indicating that MA is successfully embedded in the cavity of HP-β-CD molecules [37].

### 3.6. Stability at 100 °C 

Figure 7 shows the release curve of MA with time in HP-β-CD/MA microcapsule samples heated at 100 °C. Under this condition, all the unembedded MA volatilized in about 40 min. At the same time, the release rate of MA in microcapsules first increased with time and then tended to be stable, and reached about 11% when the heating time was 20 min. This may be due to the volatilization of water and a small amount of unembedded MA in the HP-β-CD/MA microcapsules. After 20 min, the release rate slowed down gradually because there were hydrogen bonds and van der Waals force interactions between HP-β-CD and MA, forming stable inclusions, resulting in the slow release of the embedded MA [29]. The thermal stability of MA was significantly improved by being embedded in HP-β-CD, and the volatilization of MA was reduced, showing an obvious slow-release effect.

### 3.7. Stability at Room Temperature

Figure 8 shows the change curve of the retention rate of MA and HP-β-CD/MA microcapsules with time at room temperature of 25 °C and humidity of 60%. As can be seen from the figure, the retention rate of MA in microcapsules decreased by about 10% within 7 days, while the content of unembedded MA decreased by about 35% within 7 days. The results showed that the microencapsulation of MA by HP-β-CD was able to significantly slow down its volatilization and improve storage stability.

## 4. Conclusions

In this study, HP-β-CD/MA microcapsules were prepared to improve the solubility and stability of MA. A gas chromatography method was developed to determine the embedding rate of HP-β-CD/MA microcapsules. The total content of MA in microcapsule samples was determined with ethanol as solvent. The average recovery was 99.8%, and RSD was 1.3%. The unembedded content of MA on the surface of microcapsules was determined with ethyl acetate as solvent. The average recovery was 100.3%, and RSD was 0.82%. Compared with the standard method, this method is more sensitive and accurate. According to the test results, HP-β-CD has an excellent embedding effect on MA, and the embedding rate was 96.3%.

According to the thermodynamic parameters determined via the phase solubility method, the embedding process of HP-β-CD and MA is a spontaneous exothermic reaction with enthalpy as the main driving force. NMR indicated that the chemical shifts of H-3 and H-5 in the cavity of HP-β-CD was −0.002 and −0.021, and the chemical shifts of C_1_-H, C_2,6_-H, C_3,7_-H, C_4,5_-H and C_10_-H of MA were −0.003, −0.001, −0.004, 0.001 and −0.002, respectively. Based on the NMR results and molecular structure, it can be concluded that MA was mainly embedded in the cavity of HP-β-CD.

DSC and stability tests at room temperature and 100 °C showed that HP-β-CD could play an excellent protective effect on MA, slow down the volatilization of MA, significantly improve the stability of MA and overcome the volatile and unstable defects of MA. The development of HP-β-CD/MA microcapsules is remarkably helpful for the use, storage and processing of MA, which could have promising applications in the medicine and food fields.

## Figures and Tables

**Figure 1 pharmaceutics-15-01979-f001:**
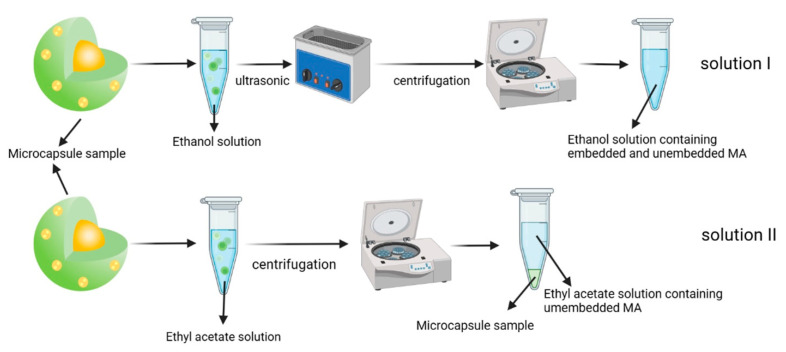
Experimental scheme of the MA detection process.

**Figure 2 pharmaceutics-15-01979-f002:**
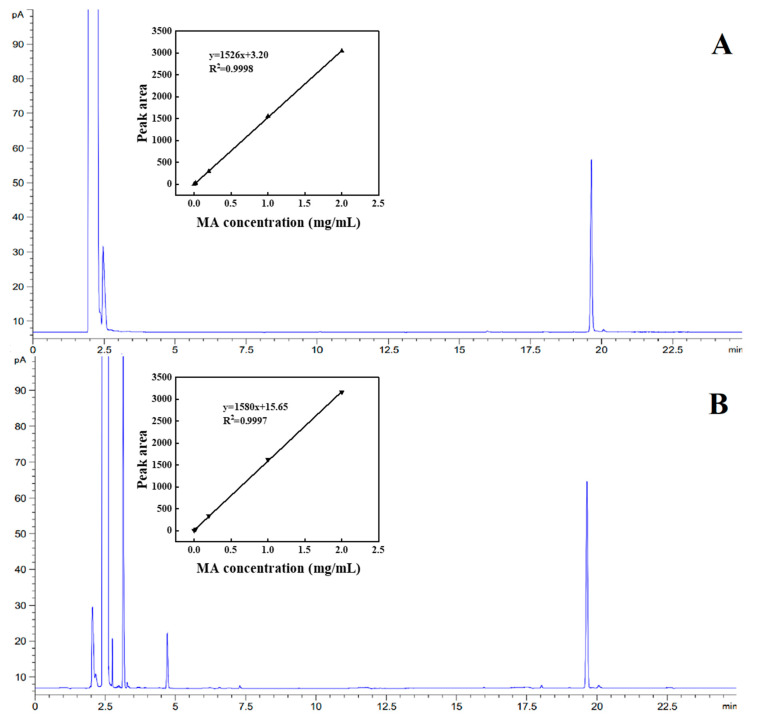
Gas chromatograms of MA with (**A**) ethanol and (**B**) ethyl acetate as solvents. Inset: standard curves of MA concentration and peak area with ethanol and ethyl acetate as solvents.

**Figure 3 pharmaceutics-15-01979-f003:**
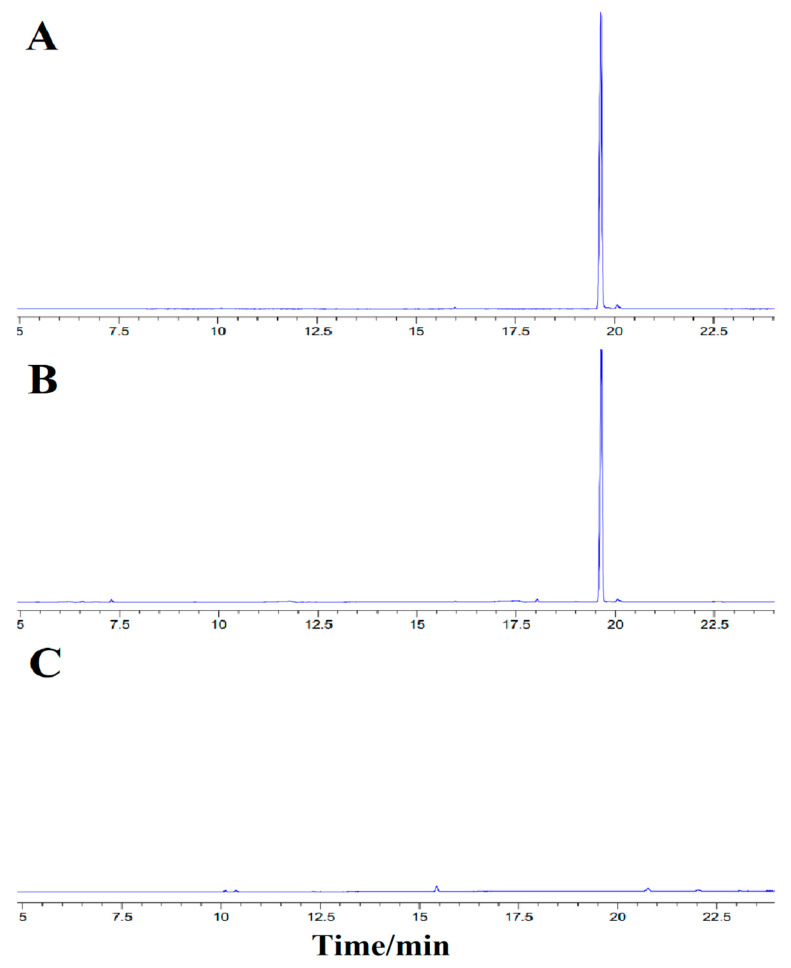
Gas chromatograms of (**A**) MA, (**B**) HP-β-CD /MA microcapsules and (**C**) HP-β-CD.

**Figure 4 pharmaceutics-15-01979-f004:**
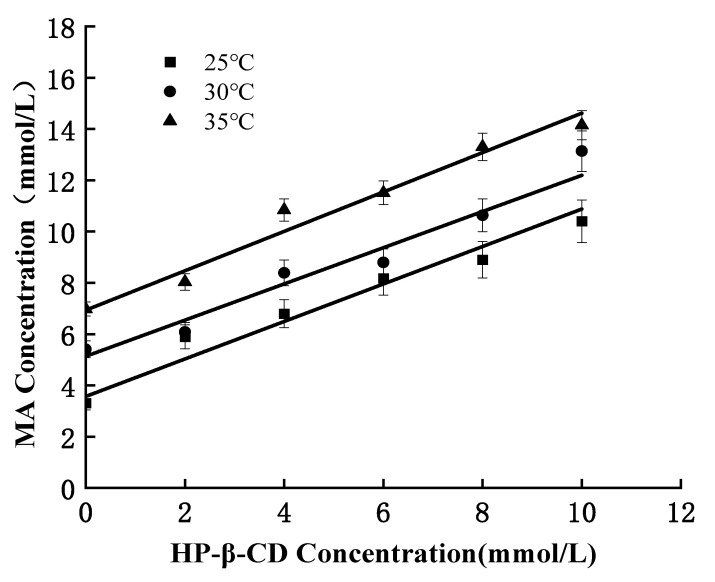
Phase dissolution diagram of HP-β-CD and MA at different temperatures.

**Figure 5 pharmaceutics-15-01979-f005:**
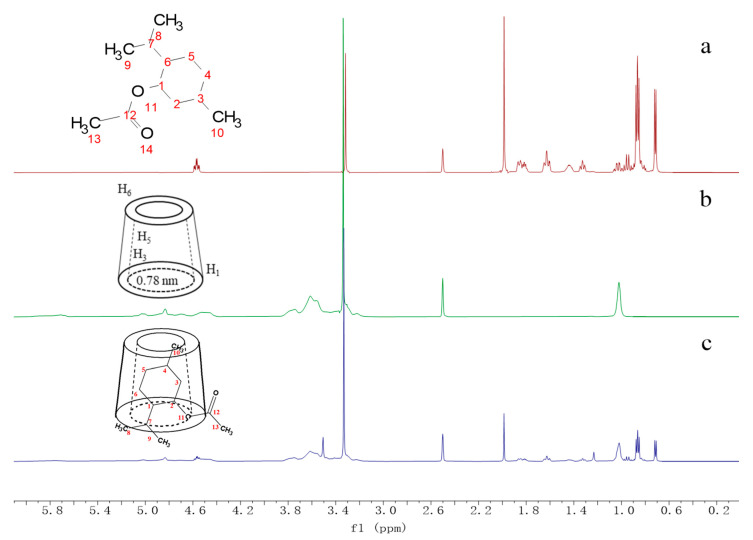
^1^H NMR spectra of MA (**a**), HP-β-CD (**b**) and HP-β-CD/MA microcapsules (**c**). Inset: proposed structure of the MA, HP-β-CD and HP-β-CD/MA microcapsule.

**Figure 6 pharmaceutics-15-01979-f006:**
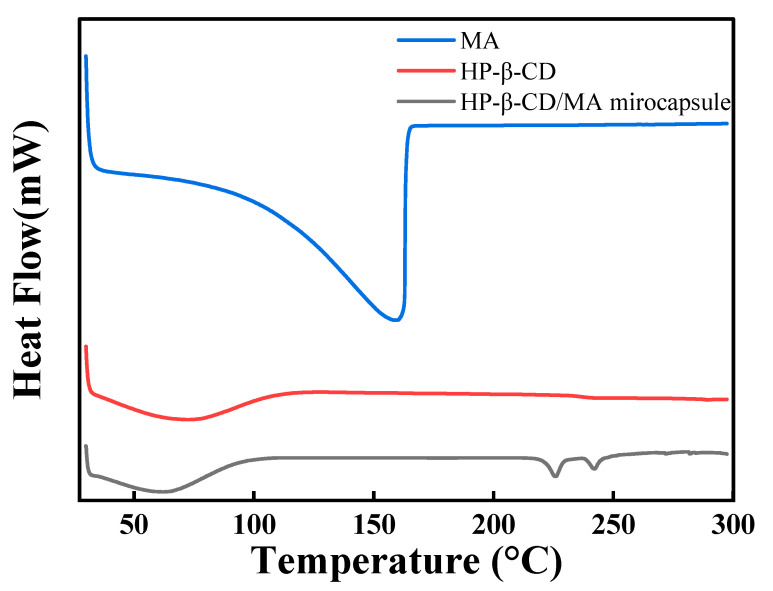
DSC diagram of MA, HP-β-CD and HP-β-CD/MA microcapsules.

**Figure 7 pharmaceutics-15-01979-f007:**
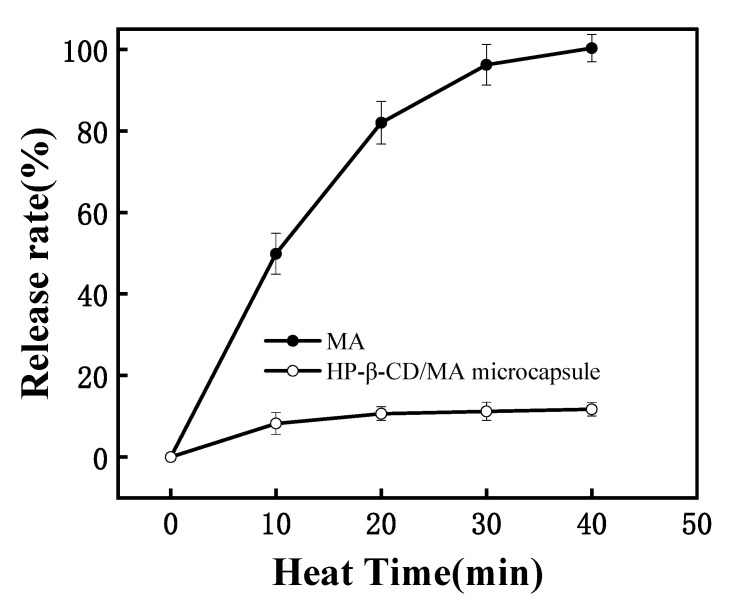
The curve of release rate of MA and HP-β-CD/MA microcapsules with time at 100 °C.

**Figure 8 pharmaceutics-15-01979-f008:**
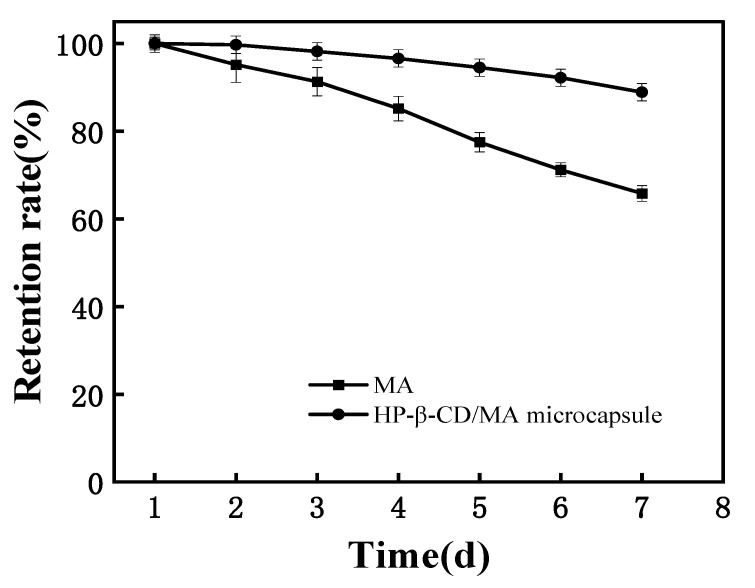
Retention curve of MA and HP-β-CD/MA microcapsules with time at temperature of 25 °C and humidity of 60%.

**Table 1 pharmaceutics-15-01979-t001:** Linear regression equation and stability constant for HP-β-CD and MA at different temperatures.

Temperature/°C	Regression Equation	R^2^	a	b	K (L/mol)
25	y = 0.7309x + 3.5710	0.9761	0.7309	3.5710	760.60
30	y = 0.7526x + 4.9818	0.9631	0.7526	4.9818	610.63
35	y = 0.7684x + 6.9345	0.9743	0.7684	6.9345	478.45

**Table 2 pharmaceutics-15-01979-t002:** Thermodynamic parameters of the embedding reaction of HP-β-CD with MA.

Temperature/°C	K/(L/mol)	ΔH (KJ/mol)	ΔS (J/mol·K)	ΔG (KJ/mol)
25	760.60	−33.016	−55.840	−16.256
30	610.63	−33.016	−55.840	−15.976
35	478.45	−33.016	−55.840	−15.697

## Data Availability

Not applicable.

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
