# Peer review of "Preparation and Embedding Characterization of Hydroxypropyl-β-cyclodextrin/Menthyl Acetate Microcapsules with Enhanced Stability"

_pharmaceutics, 2023, doi:10.3390/pharmaceutics15071979_

Round 1

Reviewer 1 Report

Interesting article. The authors designed microparticles in which Hydroxypropyl-β-cyclodextrin was used as a shell, and Menthyl acetate was the active ingredient. The work uses modern methodology and instruments. The work has a good theoretical basis. However, the article needs to be significantly improved.

There are many repetitions in the post. The Results section contains many repetitions from the Materials and Methods section.

The division of the Results section into subsections is not always justified. Lack of discussion of results. All discussion is reduced to the arrangement of references to the literature. Very confusing description in subsections 3.1 - 3.3. In the results section, the names of normative documents are given. For a better understanding, a research scheme should be given. It is not clear how many methods were compared to identify MA. How many solutions were used? How were they prepared?

Line. 72 – 74 Why is there information from the materials and methods section in the introduction section?

Line 143. What is the intersection of the regression equation with, or the intersection of the fitting line with something?

Line 194 - 195 repeat from Materials and Methods

Lines 195 - 196 It is not clear how many methods were used to determine the MA. An experiment scheme with a description should be added to the materials and methods.

Line 196 – 197 “…good linear relationship in the concentration range of 0.002~2 mg/mL….” Line 197 – 198 “The detection limit and quantitative limit were 0.83 μg/mL and 2.52 μg/mL, respectively….” Why was quantitative limit 2.52 μg/mL, if the linear relationship persists up to 2 mg/mL?

All standards names should be in the Reference section. In the materials and methods section, provide a brief description and provide a link to the normative document. The results section should not contain a description of the methodology. The data obtained using the standard method should be designated "Control".

What are regression equations for? They are not used in further work. Add a drawing with a calibration curve to the chromatograms, write the accuracy of the approximation above each curve and make a description.

What is the reason for such a discrepancy between the results obtained by the standard method and your results. Should be discussed.

Item 3.1.2 What is it? Is this a method description? Where are the results?

Item 3.1.3 What is it? Is this a method description? Where are the results?

Item 3.2 Remove figure 2. Shorten the description of the chromatography. If the microcapsules were obtained under the same conditions, then paragraph 3.2 does not make sense. It is necessary to give a description of the obtained microparticles at the beginning of the Results section. If the microparticles were obtained by varying the parameters, then the results should be reported.

In table 1, the coefficients "a" should be given; "b" and K. R2 should be placed on the chart.

A thermogram of pure HP-β-CD should be added to Figure 5.

The conclusion section should contain conclusions on the results of the work, supported by specific figures, and not a brief retelling of the chapter's results.

It needs to be cleaned of unnecessary information, with concretization.

Reviewer 2 Report

In the peasant manuscript entitled "Preparation and embedding characterization of hydroxypropyl-2 β-cyclodextrin/menthyl acetate microcapsules with enhanced 3 stability"  the authors have endeavoured to improve the stability of Menthy acetate which is a primary component in peppermint. However, some queries need to clarify before publication. my specific comments are below. 

1) Preparation of Menthyl acetated B-cyclodextrin complexes has been published already  (https://doi.org/10.1007/s10847-016-0599-y) and I would like to hear from the authors' side, what is the novelty in the current work?

 2) Rather than using the term "Deta processing" use the word "statistical analysis" which is more appropriate and mentioned all the statistics that authors have used. 

3) it is suggested to the authors mention the exact temperature in the graphs also. 

Reviewer 3 Report

Dear author,

The article entitled “Preparation and embedding characterization of hydroxypropyl-β-cyclodextrin/menthyl acetate microcapsules with enhanced stability” has been intensively reviewed and evaluated. Although present study was considered an short study, there were some major and minor points that need to be revised. Hereby, I would like to present my suggestions and revisions.

Revision_1: Authors must keep the abstract to 200 words, according to the journal template.

Revision_2: The introduction must be thorough, regarding the reason for choosing this cyclodextrin. This specifics cyclodextrin is capable of altering membrane permeability and facilitating the entry of drugs into the cells. it is non-toxic and safe, as shown by this in vitro study. This aspect needs to be described.

Revision_3: I noticed that there are no references on the methods used.

I suggest it below, but it is necessary that the authors add a bibliographic reference for each method or specify that it is a method used for the first time in their laboratory.

Lines 141-143: Higuchi and Connors' theory

-        THEORY: T. Higuchi, T.; Connors, K.A. Phase solubility technique. Adv. Anal. Chem. Instrum. 1965, 4, 117–211

Revision_4: Modelling studies could provide greater clarity on the inclusion complex.

Revision_5: characterization is the main part of this work. I suggest the authors add a picture of the probable geometry of the complex.

Revision_6: development and characterization have been carefully described and conducted, but the work lacks a biological evaluation. Have the authors thought of a possible biological application? It may be important to do in vitro biocompatibility tests.

Revision_7: the authors must explain in detail the purpose of the work. Why is it necessary to increase the stability of this active ingredient? What is the goal? Authors should thoroughly discuss and support with references and possibly in vitro biological studies (Critically important request)

Round 2

Reviewer 1 Report

The authors have done a great job. The manuscript has been greatly improved and may be published.

Reviewer 3 Report

Dear authors,

for me the manuscript can be published.